# Decoupled Sequence and Structure Generation for Realistic Antibody Design

**Nayoung Kim**                                                    *nayoungkim@kaist.ac.kr*
*Korea Advanced Institute of Science and Technology (KAIST)*

**Minsu Kim**                                                        *min-su@kaist.ac.kr*
*Korea Advanced Institute of Science and Technology (KAIST)*

**Sungsoo Ahn**                                                    *sungsoo.ahn@postech.ac.kr*
*Pohang University of Science and Technology (POSTECH)*

**Jinkyoo Park**                                                    *jinkyoo.park@kaist.ac.kr*
*Korea Advanced Institute of Science and Technology (KAIST)*

**Reviewed on OpenReview:** *https://openreview.net/forum?id=CTkABQvnkm*

## Abstract

Recently, deep learning has made rapid progress in antibody design, which plays a key role in the advancement of therapeutics. A dominant paradigm is to train a model to jointly generate the antibody sequence and the structure as a candidate. However, the joint generation requires the model to generate both the discrete amino acid categories and the continuous 3D coordinates; this limits the space of possible architectures and may lead to suboptimal performance. In response, we propose an antibody sequence-structure decoupling (ASSD) framework, which separates sequence generation and structure prediction. Although our approach is simple, our idea allows the use of powerful neural architectures and demonstrates notable performance improvements. We also find that the widely used non-autoregressive generators promote sequences with overly repeating tokens. Such sequences are both out-of-distribution and prone to undesirable developability properties that can trigger harmful immune responses in patients. To resolve this, we introduce a composition-based objective that allows an efficient trade-off between high performance and low token repetition. ASSD shows improved performance in various antibody design experiments, while the composition-based objective successfully mitigates token repetition of non-autoregressive models.

## 1 Introduction

Antibodies are Y-shaped proteins that detect and neutralize the disease-causing agents. Due to high specificity and binding affinity, they are recognized as one of the most promising drug modalities (Mullard, 2022). Historically, antibody development relied on experimental and physics-based computational approaches, which are laborious and time-consuming (Kim et al., 2023). In response, there is a growing trend to integrate deep learning-based approaches for antibody development.

Early deep generative models (Liu et al., 2020; Saka et al., 2021; Akbar et al., 2022) focused only on designing the antibody sequences, neglecting the importance of antibody structure in functionality (Martinkus et al., 2024). To address this, Jin et al. (2021) proposed the sequence-structure co-design to generate both antibody sequences and structures. Many works made notable progress, including Jin et al. (2021); Verma et al. (2023); Kong et al. (2022); Luo et al. (2022); Wu & Li (2024). Such models are based on the joint prediction of sequence and structure with a common architecture for both tasks. However, this restricts task-specific optimization of model architectures, potentially hindering higher sequence-structure modeling performance.

Additionally, recent works (Melnyk et al., 2023; Kong et al., 2022; Verma et al., 2023; Kong et al., 2023) show that sequence generation with non-autoregressive models[1] achieves faster inference speed and better performance than the autoregressive counterparts, with its efficacy even extending to protein design (Gao et al., 2022; Zheng et al., 2023). However, we find that such models generate sequences that are disproportionately dominated by frequently occurring amino acid type (e.g., the natural sequence `ARMGSDYDVWFDY` versus non-autoregressive prediction `TRYYYYYYYYYDY`). Such sequences are out-of-distribution and prone to undesirable developability properties that can trigger harmful immune responses in patients (Jin et al., 2021; Wang et al., 2009).

**Contribution.** In this paper, we propose antibody sequence-structure decoupling (ASSD), a general framework for constructing a highly performant antibody sequence-structure co-design model. Our idea is to leverage Anfinsen's dogma, which states that the native conformation of a protein is determined by amino acid sequence (Anfinsen, 1973). To this end, instead of joint prediction, we decouple the sequence-structure design into two steps: a sequence design step followed by a structure prediction step. This decoupling approach is desirable since it allows the suitable choice of model architectures for each task and enables the use of large-scale sequence databases (without structure information) during the sequence design step. Despite the simplicity, such a decoupling of sequence-structure co-design has been overlooked in previous works.

We also propose a composition-based objective for antibody sequence generation that resolves the token repetition problem of non-autoregressive models. Specifically, our key idea is to augment the MLE objective with the maximization of similarity between amino acid compositions of generated and target sequences. To this end, we use REINFORCE trick (Williams, 1992; Fu et al., 2015) with respect to the composition vector which accounts for the rate of amino acids appearing in the sequences. Our idea is to let the model implicitly learn the dependency between the residues within a sequence, effectively preventing excessive repetition of any single amino acid.

Our experiments demonstrate that our sequence-structure decoupling approach improves performance in various antibody design experiments, while our training algorithm effectively prevents excessive token repetitions. Notably, our approach establishes a Pareto frontier over other non-autoregressive antibody design models, indicating optimal trade-offs between high sequence modeling capacity and low token repetition. Additionally, we demonstrate that our training algorithm can be generalized to protein design.

## 2 Related work

**Antibody design.** Many pioneering works on antibody design predicts only 1D sequences using CNN and LSTM (Liu et al., 2020; Saka et al., 2021; Akbar et al., 2022). Recently, Melnyk et al. (2023) improved 1D sequence modeling performance by repurposing a pre-trained Transformer-based English language model for sequence infilling. Frey et al. (2023) also demonstrated the potential of energy/score-based models for antibody sequence design with a walk-jump sampling scheme.

Instead of designing only 1D sequences, Jin et al. (2021) proposes sequence-structure co-design, which involves both 1D sequence and 3D structure infilling of CDRs. Specifically, Jin et al. (2021) represents the antibody-antigen complex with an E(3)-invariant graph and predicts the sequence in an autoregressive fashion. On the contrary, Kong et al. (2022) constructs an E(3)-equivariant graph to fully capture the 3D geometry and predicts sequence by a full-shot decoding scheme to speed up inference. Verma et al. (2023) formulates a coupled neural ODE system over the antibody nodes and further increases inference speed with a single round of full-shot decoding. Wu & Li (2024) introduces a four-level training strategy with various protein/antibody databases. Unlike deterministic GNN-based methods, Luo et al. (2022) and Martinkus et al. (2024) adopt diffusion probabilistic models to generate structures stochastically. Notably, Frey et al. (2023); Martinkus et al. (2024) supports their approach with laboratory experiments.

Previous approaches predict the sequence and structure jointly, except Martinkus et al. (2024) which introduces a structure-to-sequence decoupling strategy. However, this approach constrains itself to the Structural Antibody Database (SAbDab) (Dunbar et al., 2014), which is orders of magnitude smaller than sequence

---

[1]In this paper, we limit our focus to one- or few-shot non-autoregressive models.

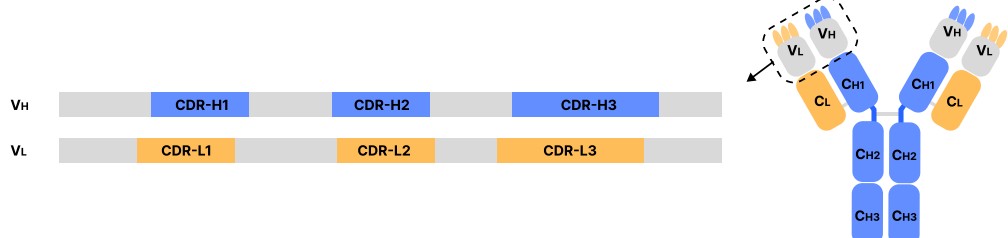

Figure 1: **Schematic structure of an antibody.** An antibody consists of a pair of heavy and light chains, each containing a variable region and constant regions. The variable region consists of three complementarity-determining regions (CDRs) and its complement called the framework regions. We aim to design heavy chain CDRs, which contribute the most to antibody-antigen interaction.

databases like the Observed Antibody Space (OAS) (Olsen et al., 2022). In this work, we demonstrate the superiority of sequence-to-structure decoupling over joint prediction strategy.

**Protein design.** Due to their relevance, methods in protein design can offer valuable insights into antibody design. Ingraham et al. (2019) represents the protein structure with a k-nearest neighbor graph and applies a Transformer-based encoder-decoder model to predict the sequences autoregressively. To fully capture the complex geometry within the graphs, Jing et al. (2020) introduces geometric vector perceptrons that can replace MLPs in GNNs. Tan et al. (2023) introduces an additional global module that captures the global context of the protein structure. To enable an application to a wide range of protein design problems, Dauparas et al. (2022) adopts an order-agnostic autoregressive model with random decoding orders. Instead, Gao et al. (2022) adopts a one-shot decoding scheme and drastically improves the inference speed. Recently, Zheng et al. (2023) showed state-of-the-art performance by incorporating a lightweight structural adapter into a protein language model.

Notably, Melnyk et al. (2023); Kong et al. (2022); Verma et al. (2023); Gao et al. (2022); Zheng et al. (2023) adopt non-autoregressive factorization trained with MLE objective or its slight variations. In this paper, we showcase that this approach promotes highly repetitive sequences and propose a novel training strategy to mitigate this problem while maintaining a comparable level of performance.

## 3 Preliminaries and notations

**Preliminaries.** As shown in Figure 1, an antibody consists of a pair of light chains and heavy chains. Both the light chain and heavy chain are composed of a variable region ($V_L$;$V_H$) and constant region(s) ($C_L$; $C_{H1}, C_{H2}, C_{H3}$). Each variable region could be further decomposed into three complementarity-determining regions (CDR-L1, CDR-L2, CDR-L3; CDR-H1, CDR-H2, CDR-H3) and a framework region, which is the complement of CDRs. Following previous works (Jin et al., 2021; Kong et al., 2022; Verma et al., 2023; Luo et al., 2022; Melnyk et al., 2023), we narrow the scope of our task to sequence-structure co-design of heavy chain CDRs as they contribute the most to antibody-antigen interaction.

**Notations.** We denote the amino acid vocabulary set with $\mathcal{A}$. Let $L$ be the number of residues in CDR. We denote the *ground-truth* sequence and structure of a heavy chain CDR as $\mathbf{s} = (s_1, \ldots, s_L)$ and $\mathbf{x} = (x_1, \ldots, x_L)$ and the *predicted* sequence and structure as $\hat{\mathbf{s}}$ and $\hat{\mathbf{x}}$. We denote all conditional information (e.g., framework region, antigen, initializations) as $\mathbf{c}$.

## 4 Methods

Here, we provide an overview of this section. Our goal is to train a non-autoregressive generator that can generate the sequence of amino acids and the associated 3D information conditioned on some semantic information. Motivated by Anfinsen's dogma, which states that the protein structure is determined by its amino acid sequence (Anfinsen, 1973), we propose an antibody sequence-structure decoupling (ASSD)

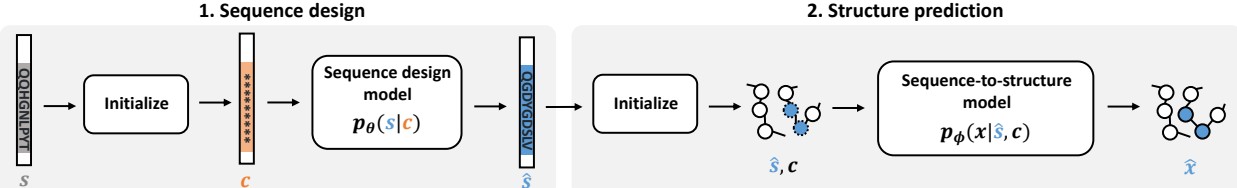

Figure 2: **Overview of antibody sequence-structure decoupling (ASSD) framework.** ASSD first designs CDR sequences with a sequence design model and then predicts the corresponding structure with a sequence-to-structure model. $\boldsymbol{s}, \boldsymbol{x}$ are the ground-truth, and $\hat{\boldsymbol{s}}, \hat{\boldsymbol{x}}$ are the predicted CDR sequence and structure. $\boldsymbol{c}$ denotes conditional information, e.g., framework region, antigen, and sequence/structure initialization.

framework that decouples sequence-structure co-design into the sequence design step (Section 4.2), followed by the structure generation step (Section 4.3). Additionally, we resolve excessive token repetition of non-autoregressive models with a composition-based sequence-level objective (Section 4.2).

## 4.1 Problem formulation & algorithm overview

**Problem statement.** We first formulate our problem for designing an antibody. Following previous works (Jin et al., 2021; Kong et al., 2022; Verma et al., 2023; Luo et al., 2022; Melnyk et al., 2023), we narrow the scope of our task to sequence-structure co-design of heavy chain complementarity-determining regions (CDRs) in the antibody as they contribute the most to antibody-antigen interaction. The heavy chain CDRs are represented by an amino acid sequence associated with 3D positions. We provide more information on the antibody structure in Figure 1.

To be specific, we let $\mathcal{A}$ denote the amino acid vocabulary set and $L$ the number of residues in the CDR. The CDR is represented by the sequence $\boldsymbol{s} = (s_1, \ldots, s_L)$ and the positional information $\boldsymbol{x} = (x_1, \ldots, x_L)$ where $s_i \in \mathcal{A}$ corresponds to a particular type of amino acid and $x_i \in \mathbb{R}^{m \times 3}$ the associated 3D positional information with $m$ backbone atoms. Our goal is to train a generator $p_\theta(\boldsymbol{s}, \boldsymbol{x} | \boldsymbol{c})$ for the amino acid sequence $\boldsymbol{s}$ and the positional information $\boldsymbol{x}$, given the conditional information $\boldsymbol{c}$, e.g., the framework region, antigen, and initialization.

**Algorithm overview.** In ASSD, we parameterize the generator $p_{\theta,\phi}(\boldsymbol{s}, \boldsymbol{x} | \boldsymbol{c})$ as a sequential composition of sequence design module $p_\theta(\boldsymbol{s} | \boldsymbol{c})$ and the structure prediction module $p_\phi(\boldsymbol{x} | \boldsymbol{s}, \boldsymbol{c})$, i.e., we let $p_{\theta,\phi}(\boldsymbol{s}, \boldsymbol{x} | \boldsymbol{c}) = p_\theta(\boldsymbol{s} | \boldsymbol{c}) p_\phi(\boldsymbol{x} | \boldsymbol{s}, \boldsymbol{c})$. This decoupling is motivated by a well-known postulate in molecular biology, called Anfinsen's dogma, which states that the native conformation of a protein is determined solely by its amino acid sequence (Anfinsen, 1973). We provide an overview of our algorithm in Figure 2.

Additionally, we consider a non-autoregressive sequence generative model where the amino acids are predicted independently from each other, i.e., $p_\theta(\boldsymbol{s} | \boldsymbol{c}) = \prod_{i=1}^{L} p_\theta(s_i | \boldsymbol{c})$, as recent works (Kong et al., 2022; 2023; Verma et al., 2023; Zheng et al., 2023) demonstrate both faster speed and better performance compared to the autoregressive counterparts. However, since the non-autoregressive models generate sequences with excessively repetitive tokens, we propose a new loss function to prevent this.

## 4.2 Sequence design

In this section, we introduce the training algorithm and implementation details for a sequence-only antibody design model $p_\theta(\boldsymbol{s} | \boldsymbol{c})$. Our insight is that the repetitive tokens from the generator stem from the mode-covering behavior of MLEs, i.e., each amino acid prediction aims for an average of different modes, i.e., the most frequent element. To overcome this limitation, we augment MLE training with a new mode-seeking training objective based on the compositional similarity of protein sequences that mitigates the repetitive token problem.

**Intuitions for repetitive sequences.** We are motivated by the observation that non-autoregressive sequence generators exhibit the degenerate behavior of generating excessive token repetitions, e.g., $s = \text{ARGYYYYYYY}$ where $\text{A}, \text{R}, \text{G}, \text{Y} \in \mathcal{A}$ are the possible types of amino acids. However, this is critical evidence that the generator fails to learn the underlying dataset – i.e., the generated sequences are out-of-distribution. Also, overly repetitive sequences should be avoided since they may cause developability issues such as aggregation that trigger harmful immune responses in patients (Jin et al., 2021; Wang et al., 2009).

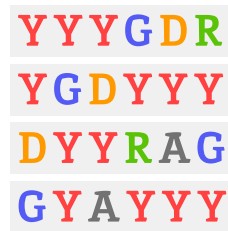

To better illustrate this problem, consider a hypothetical example with five possible amino acids $\{\text{A}, \text{R}, \text{G}, \text{Y}, \text{D}\}$. The training dataset consists of the same conditional information $c$ and sequences $s$ in Figure 3. The MLE objective of a non-autoregressive model $p_\theta(s|c) = \prod_{i=1}^{L} p_\theta(s_i|c)$ is given by

Figure 3: Example to illustrate token repetition problem.

$$\mathcal{L}_{\text{MLE}}(\theta) = \sum_{n=1}^{N} \sum_{i=1}^{L} \log p_\theta(s_i^{(n)}|c) = \sum_{i=1}^{L} \sum_{n=1}^{N} \ell_{\text{ce}}(s_i^{(n)}, \boldsymbol{p}_i),$$

where $\boldsymbol{p} = (\boldsymbol{p}_1, \ldots, \boldsymbol{p}_L) = f_\theta(c)$ is the head output distribution. If we train the model with this objective, it assigns the highest likelihood to the sequence $\text{YYYYYY}$ which is overly repetitive (proof in Appendix A). Such a phenomenon indeed occurs for antibody or protein sequences, where in certain regions, some residues have a higher frequency than others (Nikula et al., 1995; Shi et al., 2014).

**Loss function.** We train our sequence-only antibody design model $p_\theta(s|c)$ on the loss function $\mathcal{L}_{\text{seq}}(\theta) = \mathcal{L}_{\text{NLL}}(\theta) + \alpha \mathcal{L}_{\text{comp}}(\theta)$, where $\mathcal{L}_{\text{NLL}}(\theta)$ is the negative log-likelihood (NLL) objective, $\mathcal{L}_{\text{comp}}$ is the proposed composition-based loss function that regularize the repetitive tokens, and $\alpha > 0$ is some hyperparameter. In particular, given a sample $(s, c)$ from the dataset, the loss function is defined as follows:

$$\mathcal{L}_{\text{NLL}}(\theta) = -\log p_\theta(s|c), \quad \mathcal{L}_{\text{comp}}(\theta) = \mathbb{E}_{\hat{s} \sim p_\theta(\cdot|c)}[d(s, \hat{s})],$$

where $d(s, \hat{s})$ is any sequence-level metric between input amino acid sequence $s$ and the reconstructed amino acid sequence $\hat{s}$. Following the existing NLP literature (Mathur et al., 2019; Kim et al., 2021; Zhang et al., 2019), we propose to use the cosine similarity defined as follows:

$$d(s, \hat{s}) = -\text{CosineSimilarity}(\boldsymbol{y}, \hat{\boldsymbol{y}}) = -\frac{\boldsymbol{y}^\top \hat{\boldsymbol{y}}}{\|\boldsymbol{y}\|\|\hat{\boldsymbol{y}}\|},$$

where $\boldsymbol{y}, \hat{\boldsymbol{y}}$ is the composition vector of the sequence $s, \hat{s}$ that counts the occurrence of each amino acid type in the sequence. For example, $y_j = \sum_{i=1}^{L} \mathbf{1}_j(s_i)$ where $\mathbf{1}_j$ is a binary indicator that has the value of one if the amino acid $s_i$ is the $j$-th type in the vocabulary $\mathcal{A}$ and zero otherwise.

**Training with REINFORCE.** To train our generator, we use the REINFORCE trick (Williams, 1992; Fu et al., 2015) with a baseline $b = \mathbb{E}_{(s,c) \sim \mathcal{B}}[d(s, \hat{s})]$ to reduce the variance of the gradients (Appendix B). This results in the following training objective:

$$\mathcal{L}_{\text{seq}}(\theta, \mathcal{B}) = \mathbb{E}_{(s,c) \sim \mathcal{B}} \left[ -\sum_{i=1}^{L} \log p_\theta(s_i|c) + \alpha \cdot [d(s, \hat{s}) - b] \log p_\theta(\hat{s}|c) \right]. \tag{1}$$

We describe the full algorithm in Algorithm 1.

**Implementation.** Our sequence-structure decoupling approach enables the use of a large-scale sequence database during the sequence design step. We implicitly exploit this advantage by adopting a protein language model (pLM) as the sequence generator. In particular, we choose ESM2-650M (Lin et al., 2023) as our starting point for training and use the LoRA fine-tuning strategy (Hu et al., 2021) (details in Appendix C).

---

**Algorithm 1** Training sequence design model

---

**Require:** Antibody sequence dataset $\mathcal{D} = \{(\boldsymbol{s}, \boldsymbol{c})\}$, antibody sequence generator $p_\theta(\boldsymbol{s}|\boldsymbol{c})$.

  **repeat**
     Sample $\mathcal{B} = \{(\boldsymbol{s}^{(i)}, \boldsymbol{c}^{(i)})\}_{i=1}^M \sim \mathcal{D}$.
     **for** $i = 1, \ldots, M$ **do**
        Sample $\hat{\boldsymbol{s}}^{(i)} \sim p_\theta(\boldsymbol{s}|\boldsymbol{c}^{(i)})$ and compute $d(\boldsymbol{s}^{(i)}, \hat{\boldsymbol{s}}^{(i)})$.
     **end for**
     Compute $b = \frac{1}{M} \sum_{i=1}^M d(\boldsymbol{s}^{(i)}, \hat{\boldsymbol{s}}^{(i)})$ and update $\theta \leftarrow \arg\min \mathcal{L}_{\text{seq}}(\theta, \mathcal{B})$ as in Equation (1).
  **until** converged

---

**Algorithm 2** Training structure prediction model

---

**Require:** A trained antibody sequence generator $p_\theta(\boldsymbol{s}|\boldsymbol{c})$, a sequence-to-structure model $p_\phi(\boldsymbol{x}|\boldsymbol{s}, \boldsymbol{c})$, and antibody sequence-structure dataset $\mathcal{D} = \{(\boldsymbol{s}, \boldsymbol{x}, \boldsymbol{c})\}$.

  **repeat**
     Sample $\mathcal{B} = \{(\boldsymbol{s}^{(i)}, \boldsymbol{x}^{(i)}, \boldsymbol{c}^{(i)})\}_{i=1}^M \sim \mathcal{D}$.
     **for** $i = 1, \ldots, M$ **do**
        Sample $\hat{\boldsymbol{s}}^{(i)} \sim p_\theta(\boldsymbol{s}|\boldsymbol{c}^{(i)})$ and predict $\hat{\boldsymbol{x}}^{(i)} \sim p_\phi(\boldsymbol{x}|\hat{\boldsymbol{s}}^{(i)}, \boldsymbol{c}^{(i)})$.
     **end for**
     Update $\phi \leftarrow \arg\min \mathcal{L}_{\text{struct}}(\phi, \mathcal{B})$ as in Equation (2).
  **until** converged

---

### 4.3 Structure prediction

Here, we describe our sequence-to-structure design model $p_\phi(\boldsymbol{x}|\boldsymbol{s}, \boldsymbol{c})$. We train the sequence-to-structure model based on the following loss function:

$$\mathcal{L}_{\text{struct}}(\phi, \mathcal{B}) = \mathbb{E}_{(\boldsymbol{s}, \boldsymbol{x}, \boldsymbol{c}) \sim \mathcal{B}} \mathbb{E}_{\hat{\boldsymbol{s}} \sim p_\theta(\cdot|\boldsymbol{c})} \left[ \frac{1}{L} \sum_{i=1}^L \ell_{\text{huber}}(x_i, \hat{x}_i) \right], \tag{2}$$

where $\ell_{\text{huber}}$ is the Huber loss to avoid numerical stability (Jin et al., 2021; Kong et al., 2022), $\hat{\boldsymbol{x}} = (\hat{x}_1, \ldots, \hat{x}_L) \sim p_\phi(\cdot|\hat{\boldsymbol{s}}, \boldsymbol{c})$ is the predicted structure given the predicted sequence $\hat{\boldsymbol{s}}$, and $p_\theta(\cdot|\boldsymbol{c})$ is the fixed sequence generative model trained using the loss function proposed in Section 4.2. Note that we train the model to predict the structure from generated antibody sequence $\hat{\boldsymbol{s}}$ instead of the ground-truth sequence from the dataset. This is to prevent poor generalization performance during inference, caused by the exposure bias (Arora et al., 2022; Bengio et al., 2015; Ranzato et al., 2016).

Exposure bias may occur since during training, the model conditions on ground-truth sequence, while during inference, it conditions on its sequence predictions. This discrepancy leads to error accumulation, where initial mistakes propagate and compound, causing the generated sequence to deviate significantly from realistic outputs. Consequently, using ground-truth sequence for training may severely undermine the model robustness (Arora et al., 2022; Bengio et al., 2015; Ranzato et al., 2016). We empirically demonstrate the benefit of using the generated sequence as input to the structure prediction model in Appendix D.

While one could consider any graph neural network (GNN) to generate the structural information, we adopt the architecture of MEAN (Kong et al., 2022) as our structure prediction model. In ablations, we demonstrate that our sequence-structure decoupling approach can improve the performance of any GNN-based joint sequence-structure co-design model.[2]

---

[2]We note that antibody structure prediction models such as IgFold (Ruffolo et al., 2023) cannot predict the structure of incomplete sequences like CDR and thus perform poorly under this setting.

Table 1: **AAR, RMSD, and $p_{\text{rep}}$ on CDR-H1 of SAbDab dataset.** AR denotes autoregressive models and NAR denotes non-autoregressive models. The best and suboptimal are bolded and underlined, respectively, with $p_{\text{rep}}$ restricted to non-autoregressive models. ASSD (MLE) achieves the best performance, while ASSD (MLE + RL) reduces $p_{\text{rep}}$ with comparable performance.

| | Models | CDR-H1 | | |
| --- | --- | --- | --- | --- |
| | | AAR($\uparrow$) (%) | RMSD($\downarrow$) | $p_{\text{rep}}(\downarrow)$ (%) |
| AR | LSTM | 39.47±2.11 | - | 0.00±0.00 |
| | AR-GNN | 48.39±4.72 | 2.90±0.17 | 0.00±0.00 |
| | RefineGNN | 42.03±2.88 | 0.87±0.10 | 0.00±0.00 |
| NAR | MEAN | 59.32±4.71 | 0.91±0.10 | 0.29±0.49 |
| | LM-Design | 64.88±1.96 | - | 0.02±0.06 |
| | ASSD (MLE) | **69.41±2.72** | **0.80±0.09** | 0.02±0.06 |
| | ASSD (MLE + RL) | 68.71±2.25 | 0.86±0.09 | **0.00±0.00** |

## 5 Antibody design experiments

**Overview.** In this section, we evaluate our model on four antibody benchmark experiments, including the SAbDAb benchmark (Section 5.1), RAbD benchmark (Section 5.2), antibody affinity optimization (Section 5.3), and CDR-H3 design with docked templates (Section 5.4).

**Baselines.** We select LSTM, AR-GNN, and RefineGNN as the autoregressive baselines with implementations provided in Jin et al. (2021). For non-autoregressive baselines, we adopt MEAN and LM-Design adapted for antibody design. We have not included Verma et al. (2023); Wu & Li (2024); Martinkus et al. (2024); Frey et al. (2023) as baselines as their training codes are not publicly available. The training details of ASSD and baselines are provided in Appendix C.

**Metrics.** Our main sequence and structure modeling evaluation metrics are amino acid recovery (AAR) and root mean squared deviation (RMSD), respectively. AAR is defined as $\text{AAR}(\mathbf{s}, \hat{\mathbf{s}}) = \sum_{i=1}^{L} \mathbb{I}(s_i = \hat{s}_i)/L$, where $\hat{\mathbf{s}}$ is the generated CDR sequence and $\mathbf{s}$ is the ground-truth CDR sequence. RMSD is defined by comparing $C_\alpha$ coordinates of generated and ground-truth CDR structures, following Jin et al. (2021); Kong et al. (2022). Although there is a controversy about whether AAR and RMSD are suitable metrics, they are still the most widely used metrics across the antibody/protein literature to assess sequence/structure modeling abilities. Further justifications and discussions on evaluation metrics are in Section 8.

As none of the antibody sequences in the SAbDab dataset contain more than six token repetitions, we define a sequence with more than six repetitions as repetitive. Based on this definition, we develop a new metric called the percentage of repetitive tokens, $p_{\text{rep}}$. Given a model's sequence predictions $\hat{\mathcal{D}} = \{\hat{\mathbf{s}}^{(n)}\}_{n=1}^{N}$, $p_{\text{rep}} = \sum_{n=1}^{N} \mathbb{I}(\hat{\mathbf{s}}^{(n)} \text{ is repetitive}) \times 100\%/N$.

### 5.1 SAbDab benchmark

In this section, we evaluate the sequence and structure modeling performance for CDR-H1, CDR-H2, and CDR-H3 using the Structural Antibody Database (SAbDab) (Dunbar et al., 2014).

**Data.** Following Kong et al. (2022), we select 3977 IMGT-numbered (Lefranc et al., 2003) complexes from SAbDab that contain the full heavy chain, light chain, and antigen sequence and structure. We then split the complexes into train, validation, and test sets according to the CDR clusterings. Specifically, we use MMseqs2 (Steinegger & Söding, 2017) to assign antibodies with CDR sequence identity above 40% to the same cluster, where the sequence identity is computed with the BLOSUM62 substitution matrix (Henikoff & Henikoff, 1992). Then we conduct a 10-fold cross-validation by splitting the clusters into a ratio of 8:1:1 for train/valid/test sets, respectively. Detailed statistics of the 10-fold dataset splits are provided in Appendix E.

Table 2: **AAR, RMSD, and $p_{\mathbf{rep}}$ on CDR-H2 of SAbDab dataset.** ASSD (MLE) achieves the best performance, while ASSD (MLE + RL) reduces $p_{\text{rep}}$ with comparable performance.

| | Models | CDR-H2 | | |
|---|---|---|---|---|
| | | AAR($\uparrow$) (%) | RMSD($\downarrow$) | $p_{\text{rep}}(\downarrow)$ (%) |
| AR | LSTM | 30.41$\pm$2.47 | - | 0.00$\pm$0.00 |
| | AR-GNN | 38.03$\pm$2.81 | 2.30$\pm$0.16 | 0.00$\pm$0.00 |
| | RefineGNN | 32.05$\pm$2.23 | 0.79$\pm$0.06 | 0.00$\pm$0.00 |
| NAR | MEAN | 48.85$\pm$2.32 | 0.88$\pm$0.07 | 0.03$\pm$0.08 |
| | LM-Design | 55.31$\pm$3.92 | - | 0.02$\pm$0.07 |
| | ASSD (MLE) | **62.10$\pm$3.87** | **0.73$\pm$0.06** | 0.02$\pm$0.07 |
| | ASSD (MLE + RL) | 61.07$\pm$0.08 | 0.78$\pm$0.06 | **0.00$\pm$0.00** |

Table 3: **AAR, RMSD, and $p_{\mathbf{rep}}$ on CDR-H3 of SAbDab dataset.** ASSD (MLE) achieves the best performance, while ASSD (MLE + RL) reduces $p_{\text{rep}}$ with comparable performance.

| | Models | CDR-H3 | | |
|---|---|---|---|---|
| | | AAR($\uparrow$) (%) | RMSD($\downarrow$) | $p_{\text{rep}}(\downarrow)$ (%) |
| AR | LSTM | 15.82$\pm$1.63 | - | 0.00$\pm$0.00 |
| | AR-GNN | 18.72$\pm$0.82 | 3.60$\pm$0.58 | 0.03$\pm$0.07 |
| | RefineGNN | 24.44$\pm$1.94 | 2.24$\pm$0.14 | 0.11$\pm$0.13 |
| NAR | MEAN | 36.50$\pm$1.60 | 2.23$\pm$0.07 | 27.40$\pm$9.23 |
| | LM-Design | 37.58$\pm$1.16 | - | 29.40$\pm$10.01 |
| | ASSD (MLE) | **39.56$\pm$1.39** | 2.21$\pm$0.08 | 9.22$\pm$3.76 |
| | ASSD (MLE + RL) | 38.67$\pm$1.64 | **2.19$\pm$0.09** | **7.02$\pm$2.91** |

**Results.** Table 1, Table 2, and Table 3 presents our results on the SAbDab benchmark. First, we highlight that ASSD trained with the MLE objective outperforms all baselines in AAR and RMSD. However, its $p_{\text{rep}}$ is much higher than the autoregressive baselines, implying that some of its sequence designs are invalid. Training with our MLE-RL objective in Equation (1) successfully reduces token repetitions, while maintaining a similar performance. In particular, for CDR-H1 and CDR-H2, we reduced token repetition to 0; for CDR-H3, we reduced $p_{\text{rep}}$ to 7.02%, which corresponds to a 23.86% reduction in token repetition compared to the MLE-only objective.

We focus on CDR-H3 to showcase that $p_{\text{rep}}$ can be further decreased by increasing $\alpha$ in the objective function. Figure 4(a) show shows that our approach establishes a Pareto frontier over other non-autoregressive baselines in terms of $p_{\text{rep}}$ and AAR. Note that $p_{\text{rep}}$ can be reduced to a level comparable to autoregressive models while maintaining AAR above 30%, a level of performance required for CDR design models (Melnyk et al., 2023). This contrasts MEAN and LM-Design, whose $p_{\text{rep}}$ is above 25% despite the high AAR.

## 5.2 RAbD benchmark

In this section, we assess the ability to model CDR-H3 on the RosettaAntibodyDesign (RAbD) dataset, a benchmark containing 60 diverse antibody-antigen complexes curated by Adolf-Bryfogle et al. (2018).

**Data.** We use the SAbDab dataset as the train/validation set and the RAbD dataset as the test set. We eliminate antibodies in SAbDab whose CDR-H3 sequences are in the same cluster as the RAbD dataset, then split the remaining clusters into train and validation sets with a ratio of 9:1. This gives 3418 and 382 antibodies for the training and validation sets, respectively.

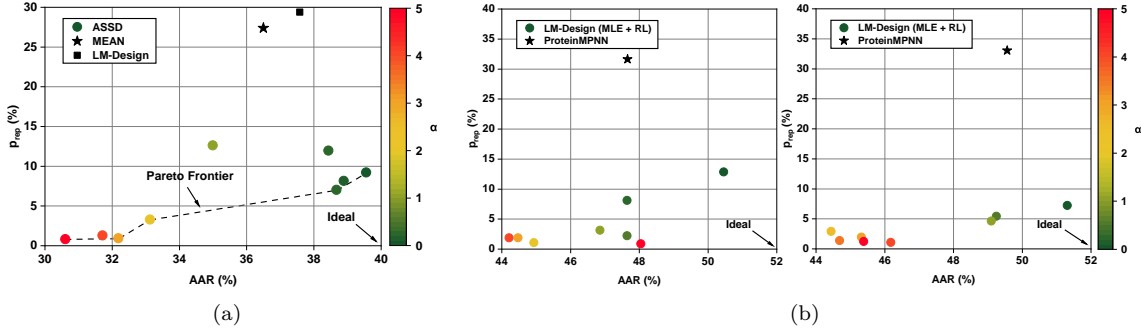

Figure 4: **Effect of $\alpha$ on AAR and $p_{\text{rep}}$.** The color bar represents the value of $\alpha$ in the MLE-RL objective. (a) Result for CDR-H3 of SAbDab benchmark. ASSD approach achieves the Pareto frontier over the non-autoregressive baselines. (b) Result for CATH 4.2 (*left*) and CATH 4.3 (*right*) benchmarks. LM-Design trained with our MLE-RL objective maintains performance comparable to ProteinMPNN, while reducing $p_{\text{rep}}$ significantly.

Table 4: **AAR, Cosim, TM-score, RMSD, and $p_{\text{rep}}$ on RAbD dataset.** AR denotes autoregressive models and NAR denotes non-autoregressive models. The best and suboptimal are bolded and underlined, respectively, with $p_{\text{rep}}$ restricted to non-autoregressive models. ASSD (MLE) and ASSD (MLE + RL) achieve the best performance, with ASSD (MLE + RL) achieving the lowest $p_{\text{rep}}$ among NAR models.

| | Models | AAR ($\uparrow$) (%) | Cosim ($\uparrow$) | TM-score ($\uparrow$) | RMSD ($\downarrow$) | $p_{\text{rep}}$ ($\downarrow$) (%) |
|---|---|---|---|---|---|---|
| AR | LSTM | 16.15 | 0.5462 | - | - | 0.00 |
| | AR-GNN | 18.85 | 0.5963 | 0.9630 | 3.56 | 0.00 |
| | RefineGNN | 26.33 | 0.5797 | 0.9648 | 1.80 | 0.00 |
| NAR | MEAN | 37.15 | 0.5832 | 0.9806 | 1.82 | 8.33 |
| | LM-Design | 38.77 | 0.5869 | - | - | 15.00 |
| | ASSD (MLE) | **40.81** | 0.6015 | 0.9825 | 1.78 | 5.00 |
| | ASSD (MLE + RL) | 40.53 | **0.6157** | **0.9830** | **1.76** | **3.33** |

**Metrics.** In addition to AAR and RMSD, we include cosine similarity (abbreviated CoSim) and TM-score (Zhang & Skolnick, 2004; Xu & Zhang, 2010) to evaluate the sequence composition similarity and global structural similarity.

**Results.** We report our results in Table 4. ASSD with MLE-only training achieves the highest AAR. While maintaining a similar level of AAR, we reduce token repetition by 33.4% with our MLE-RL objective function. Additionally, ASSD shows the highest structure modeling capacity, reflected by a high TM-score and low RMSD. This indicates the superiority of our sequence-structure decoupling approach over the joint sequence-structure co-design models like MEAN, RefineGNN, and AR-GNN.

## 5.3  Antibody affinity optimization

The goal of this task is to optimize the binding affinity of a given antibody-antigen complex by re-designing the CDR-H3 sequence and structure. Following Kong et al. (2022), we pre-train the models on the SAbDab dataset and fine-tune them with the ITA algorithm (Yang et al., 2020) on the SKEMPI V2.0 dataset (Jankauskaitė et al., 2019). To evaluate the change in binding affinity $\Delta\Delta G$, we use the official checkpoint of the oracle $f$ from Shan et al. (2022). Since $\Delta\Delta G$ evaluation requires structure, only sequence-structure co-design models are used as the baselines.

**ITA algorithm.** Following Kong et al. (2022), we adopt the ITA algorithm adjusted for continuous properties. The algorithm consists of two parts – (1) augmenting the dataset and (2) training the model on the

Table 5: **Change in binding affinity $\Delta\Delta G$ after affinity optimization on SKEMPI V2.0 dataset.** The best and suboptimal are bolded and underlined, respectively, with $p_{\text{rep}}$ restricted to non-autoregressive models. ASSD (MLE) generates re-designs with the highest binding affinities on average. ASSD (MLE + RL) achieves similar performance, while reducing $p_{\text{rep}}$ by 16.67%.

|  | Random* | AR-GNN | RefineGNN | MEAN | ASSD (MLE) | ASSD (MLE + RL) |
|---|---|---|---|---|---|---|
| $\Delta\Delta G(\downarrow)$ | +1.520 | -2.279 | -5.437 | -7.199 | **-8.454** | -8.162 |
| $p_{\text{rep}}(\downarrow)$ (%) | - | 0.000 | 3.774 | 50.943 | 11.321 | **9.434** |

Table 6: **Average binding affinity of re-designed complexes from docked templates, measured in Rosetta Energy Units.** ASSD (MLE) designs antibodies with the highest binding affinities, while ASSD (MLE + RL) achieves similar performance with $p_{\text{rep}}$ reduced by 66.7%.

|  | RefineGNN | MEAN | ASSD (MLE) | ASSD (MLE + RL) |
|---|---|---|---|---|
| Affinity ($\downarrow$) | 3036.17 | 815.32 | **624.65** | 666.90 |
| $p_{\text{rep}}(\downarrow)$ (%) | 0.00 | 11.67 | 10.00 | **3.33** |

augmented dataset. To augment the dataset $\mathcal{D}$, we select an antibody-antigen complex from $\mathcal{D}$ and generate $\mathcal{M} = 20$ candidate structures with the model. If a candidate is valid and exhibits $f < 0$, we add it to $\mathcal{D}$ and form the augmented dataset $\mathcal{Q}$. Then, we fine-tune the model on $\mathcal{Q}$. We repeat this two-step process for $T = 20$ iterations. The details of this algorithm are in Appendix F.

**Results.** As shown in Table 5, ASSD generates re-designs with the lowest binding affinities on average. By training ASSD with the MLE-RL objective (Equation (1)), we can reduce $p_{\text{rep}}$ by 16.67% while maintaining a similar level of performance. Note that this corresponds to 81.48% reduction in token repetition with better performance compared to MEAN, whose approximately half of its re-designed sequences are repetitive and thus out-of-distribution. For better comparison, we also provide the effect of random mutation from Kong et al. (2022), denoted Random*.

## 5.4 CDR-H3 design with docked templates

Here, we explore a more challenging setting where both CDR-H3 and the binding complex are unknown. The goal of this task is to generate CDR-H3 designs with higher binding affinities in this challenging setting.

**Data.** We modify the RAbD test set from Section 5.2 to create docked templates. More concretely, we first segregate each antibody-antigen complex into a CDR-H3-removed antibody and antigen, which is then used to generate 10,000 docked templates with MEGADOCK (Ohue, 2023). Among them, we include the top-10 scoring docked templates in the test set.

**Evaluation.** We prepare the best validation checkpoints from Section 5.2 and generate CDR-H3 re-design for each docked template in the test set. We then evaluate the binding affinities of the re-designed complexes with pyRosetta (Chaudhury et al., 2010) including the side-chain packing. Considering the risk of docking inaccuracy, we report the binding affinities of the top-scoring re-design for each antibody-antigen complex following Kong et al. (2022).

**Results.** Table 6 shows our results. Although ASSD trained with regular MLE objectives performs the best, about 10% of its generated sequences are overly repetitive. By training the model with the MLE-RL objective, we reduce $p_{\text{rep}}$ to 3.33%, a level comparable to the autoregressive model RefineGNN.

## 6 Protein design experiments

In this section, we demonstrate that our training algorithm is generalizable to single-chain protein design with CATH 4.2 and CATH 4.3 benchmarks. Specifically, we train LM-Design (Zheng et al., 2023) with our objective at varying levels of $\alpha$ and report the sequence modeling performance with AAR and sequence validity with $p_{\text{rep}}$. As none of the sequences in CATH 4.2 and CATH 4.3 test sets contain over ten repeated

Table 7: **Antibody benchmark results of the joint sequence-structure co-design models and their decoupled variants.** All variants use ESM2-650M as the sequence design model and thus share the same results for AAR. The decoupled variants improve both AAR and RMSD across all CDRs.

| | SAbDab | | RAbD | |
| --- | --- | --- | --- | --- |
| | AAR ($\uparrow$) (%) | RMSD ($\downarrow$) | AAR ($\uparrow$) (%) | RMSD ($\downarrow$) |
| AR-GNN | 17.47 | 3.638 | 18.81 | 3.237 |
| AR-GNN* | **41.32** | **2.807** | **40.45** | **3.199** |
| RefineGNN | 23.49 | 2.173 | 22.43 | 1.716 |
| RefineGNN* | **41.32** | **2.000** | **40.45** | **1.581** |
| MEAN | 35.39 | 2.188 | 37.15 | 1.686 |
| MEAN* | **41.32** | **2.180** | **40.45** | **1.672** |

tokens, we define a sequence with more than ten token repetitions as repetitive. We also report the result of ProteinMPNN (Dauparas et al., 2022) for comparison.

**Results.** Figure 4(b) shows our results on CATH 4.2 (*left*) and CATH 4.3 (*right*) benchmarks. Both LM-Design and ProteinMPNN, trained with their original objective, achieve high $p_{\text{rep}}$, indicating that a considerable portion of its sequence predictions are out-of-distribution. By increasing $\alpha$ in the MLE-RL objective (Equation (1)), we reduce the $p_{\text{rep}}$ while maintaining a decent level of sequence recovery. In particular, our objective allows achievement of $p_{\text{rep}} \sim 0\%$ for CATH 4.2 and $\sim 5\%$ for CATH 4.3, while maintaining AAR similar to ProteinMPNN.

## 7 Ablations

### 7.1 Effect of sequence structure decoupling

Thus far, we have validated that the sequence-structure decoupling approach with ESM2-650M and MEAN outperforms other models in antibody benchmark experiments. Here, we demonstrate that our decoupling approach can improve the performance of other joint sequence-structure co-design models (i.e., RefineGNN, AR-GNN) by revisiting the benchmarks experiments in Section 5.1 (SAbDab) and Section 5.2 (RAbD) with a focus on CDR-H3 region.

**Baselines.** We select AR-GNN, RefineGNN, and MEAN as our baselines. We also create the decoupled variants of each model by using ESM2-650M for the sequence design model and each baseline as a structure prediction model. We denote each variant with an asterisk *.

**Training.** We train the baselines with the objectives proposed in their original papers (Jin et al., 2021; Kong et al., 2022). For the decoupled variants, we train ESM2-650M with the cross-entropy objective and each structure prediction model with the same objective as the baseline. To fully reveal the benefit of our approach, we train all models to their full capacity by setting the training epochs to 9999 and patience to 10.

**Results.** Table 7 shows that the decoupled variants outperform the original joint prediction models. This demonstrates the advantage of task-specific optimization with the ASSD framework. We note that all variants use ESM2-650M as the sequence design model and thus share the same results for AAR.

### 7.2 Data leakage of ESMs

In this section, we address the potential concern regarding the influence of data leakage from protein language models on our performance. To address this concern, we repeat the experiment on the SAbDab benchmark (Section 5.1) by training the corresponding Transformer architecture from scratch.

**Results.** As shown in Table 8, Table 9, and Table 10, our approach still achieves the best results, excluding the baseline that uses ESM. This indicates that the high performance is mainly due to sequence-structure

Table 8: **AAR, RMSD, and $p_{\mathbf{rep}}$ on CDR-H1 of SAbDab dataset.** ASSD still achieves the best performance without using ESM. This suggests that its performance is mainly driven by sequence-structure decoupling rather than data leakage of ESMs.

| | Models | CDR-H1 | | |
|---|---|---|---|---|
| | | AAR($\uparrow$) (%) | RMSD($\downarrow$) | $p_{\text{rep}}(\downarrow)$ (%) |
| AR | LSTM | 39.47$\pm$2.11 | - | 0.00$\pm$0.00 |
| | AR-GNN | 48.39$\pm$4.72 | 2.90$\pm$0.17 | 0.00$\pm$0.00 |
| | RefineGNN | 42.03$\pm$2.88 | 0.87$\pm$0.10 | 0.00$\pm$0.00 |
| NAR | MEAN | 59.32$\pm$4.71 | 0.91$\pm$0.10 | 0.29$\pm$0.49 |
| | ASSD (MLE) | **68.94$\pm$2.90** | **0.86$\pm$0.19** | **0.02$\pm$0.06** |
| | ASSD (MLE + RL) | 65.23$\pm$4.59 | 0.87$\pm$0.12 | **0.02$\pm$0.05** |

Table 9: **AAR, RMSD, and $p_{\mathbf{rep}}$ on CDR-H2 of SAbDab dataset.** ASSD still achieves the best performance without using ESM. This suggests that its performance is mainly driven by sequence-structure decoupling rather than data leakage of ESMs.

| | Models | CDR-H2 | | |
|---|---|---|---|---|
| | | AAR($\uparrow$) (%) | RMSD($\downarrow$) | $p_{\text{rep}}(\downarrow)$ (%) |
| AR | LSTM | 30.41$\pm$2.47 | - | 0.00$\pm$0.00 |
| | AR-GNN | 38.03$\pm$2.81 | 2.30$\pm$0.16 | 0.00$\pm$0.00 |
| | RefineGNN | 32.05$\pm$2.23 | **0.79$\pm$0.06** | 0.00$\pm$0.00 |
| NAR | MEAN | 48.85$\pm$2.32 | 0.88$\pm$0.07 | 0.03$\pm$0.08 |
| | ASSD (MLE) | **60.39$\pm$3.53** | **0.79$\pm$0.08** | **0.02$\pm$0.07** |
| | ASSD (MLE + RL) | 60.23$\pm$4.76 | 0.83$\pm$0.07 | **0.02$\pm$0.07** |

decoupling rather than data leakage. Our findings are consistent with Wu & Li (2024) and Wang et al. (2022), which show that protein language models alone are insufficient for antibody-related tasks.

## 8 Discussions

**Conclusion.** In this paper, we proposed antibody sequence-structure decoupling (ASSD), a general framework for highly performant sequence-structure co-design models, and a composition-based objective for reducing token repetitions for non-autoregressive models. Experiments on various benchmarks demonstrate that our approach achieves high sequence-structure modeling capacity with limited token repetitions compared to previous works.

**Structure-based drug design.** A key consideration is whether using a pre-trained protein language model for sequence design overlooks essential principles of structure-based drug design by not explicitly conditioning on structural information. However, recent wet lab experiments (Hie et al., 2024) demonstrate that pre-trained protein language models can successfully design functional antibody sequences with high binding affinity, even without explicit structure modeling. Our work builds on this empirical finding, assuming that ESM has implicitly captured structural information, as evidenced in Hie et al. (2024) and Lin et al. (2023). Additionally, the proposed ASSD framework is adaptable to sequence design models that condition on structural data, suggesting a promising direction for future research.

**Limitations and future work.** A limitation of our work is using AAR and RMSD to assess the antibody design models. Both metrics cannot measure the functional properties of the designed antibodies, which are important in real-life drug discovery pipelines. However, currently, there is no standardized metric for assessing the functional properties; thus, AAR and RMSD are the most widely used metrics for antibody/protein design tasks (Jin et al., 2021; Kong et al., 2022; 2023; Verma et al., 2023; Melnyk et al., 2023; Luo et al., 2022;

Table 10: **AAR, RMSD, and $p_{\rm rep}$ on CDR-H3 of SAbDab dataset.** ASSD still achieves the best performance without using ESM. This suggests that its performance is mainly driven by sequence-structure decoupling rather than data leakage of ESMs.

| | Models | CDR-H3 | | |
|---|---|---|---|---|
| | | AAR($\uparrow$) (%) | RMSD($\downarrow$) | $p_{\rm rep}(\downarrow)$ (%) |
| AR | LSTM | 15.82±1.63 | - | 0.00±0.00 |
| | AR-GNN | 18.72±0.82 | 3.60±0.58 | 0.03±0.07 |
| | RefineGNN | 24.44±1.94 | 2.24±0.14 | 0.11±0.13 |
| NAR | MEAN | 36.50±1.60 | **2.23±0.07** | 27.40±9.23 |
| | ASSD (MLE) | 37.44±1.48 | **2.23±0.07** | 25.32±12.2 |
| | ASSD (MLE + RL) | **37.84±1.21** | 2.24±0.10 | **21.57±14.78** |

Wu & Li, 2024). A related limitation is that the amino acid composition used to construct our composition-based objective is not a functional property. Therefore, promising future research would be developing and training antibody design models based on functional property evaluators.

For discussions on why non-autoregressive models may outperform autoregressive models biological tasks, we refer readers to Appendix H.

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

## A    Proof for hypothetical example

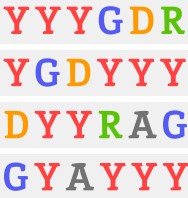

Figure 5: Sequence dataset for hypothetical example in Section 4.1.

We here provide proof for the hypothetical example in Section 4.1. Following from Section 4.1, the MLE objective is given by:

$$\mathcal{L}_{\mathrm{MLE}}(\theta) = \sum_{i=1}^{L} \sum_{n=1}^{N} \ell_{\mathrm{ce}}(s_i^{(n)}, \boldsymbol{p}_i)$$

where $\boldsymbol{p} = (\boldsymbol{p}_1, \ldots, \boldsymbol{p}_L) = f_\theta(\boldsymbol{c})$ is the head output distributions. We can re-parameterize the objective for each $\boldsymbol{p}_i$ as:

$$\mathcal{L}_{\mathrm{MLE}}(\boldsymbol{p}_i) = \sum_{n=1}^{N} \ell_{\mathrm{ce}}(s_i^{(n)}, \boldsymbol{p}_i)$$

where $\sum \boldsymbol{p}_i = 1$ is the constraint. We then compute the optimal $\boldsymbol{p}_i^*$ subject to this constraint with Sequential Least Squares Programming (SLSQP):

$$1\boldsymbol{p}_1^* = \left(0, \frac{\mathbf{2}}{\mathbf{4}}, \frac{1}{4}, \frac{1}{4}, 0\right), \quad \boldsymbol{p}_2^* = \left(0, \frac{\mathbf{3}}{\mathbf{4}}, \frac{1}{4}, 0, 0\right), \quad \boldsymbol{p}_3^* = \left(\frac{1}{4}, \frac{\mathbf{2}}{\mathbf{4}}, 0, \frac{1}{4}, 0\right)$$

$$\boldsymbol{p}_4^* = \left(0, \frac{\mathbf{2}}{\mathbf{4}}, \frac{1}{4}, 0, \frac{1}{4}\right), \quad \boldsymbol{p}_5^* = \left(\frac{1}{4}, \frac{\mathbf{2}}{\mathbf{4}}, 0, \frac{1}{4}, 0\right), \quad \boldsymbol{p}_6^* = \left(0, \frac{\mathbf{2}}{\mathbf{4}}, \frac{1}{4}, 0, \frac{1}{4}\right)$$

where $\boldsymbol{p}_i = (p_A, p_Y, p_G, p_D, p_R)$ respectively. Since $p_{\theta^*}(\boldsymbol{s}|\boldsymbol{c}) = \prod_{i=1}^{6} \mathrm{Cat}(s_i|\boldsymbol{p}_i^*)$, the sequence with highest likelihood is YYYYYY.

## B Derivation of objective $\mathcal{L}_{\mathbf{comp}}(\theta)$

We here derive a differentiable objective function of $\mathcal{L}_{\mathrm{comp}}(\theta) = \mathbb{E}_{\hat{s} \sim p_\theta(\hat{s}|c)}[d(s, \hat{s})]$ using the REINFORCE estimator (Fu et al., 2015). Consider a sample $(s, c)$ from the dataset.

$$
\begin{aligned}
\nabla_\theta \mathcal{L}_{\mathrm{comp}}(\theta) &= \nabla_\theta \mathbb{E}_{\hat{s} \sim p_\theta(\hat{s}|c)}[d(s, \hat{s})] \\
&= \nabla_\theta \sum_{\hat{s}} p_\theta(\hat{s}|c)d(s, \hat{s}) \\
&= \sum_{\hat{s}} \left[\nabla_\theta p_\theta(\hat{s}|c)\right] d(s, \hat{s}) \\
&= \sum_{\hat{s}} p_\theta(\hat{s}|c)\nabla_\theta \log p_\theta(\hat{s}|c)d(s, \hat{s}) \\
&= \mathbb{E}_{p_\theta(\hat{s}|c)}[d(s, \hat{s})\nabla_\theta \log p_\theta(\hat{s}|c)] \\
&= \mathbb{E}_{p_\theta(\hat{s}|c)}[(d(s, \hat{s}) - b)\nabla_\theta \log p_\theta(\hat{s}|c) + b\nabla_\theta \log p_\theta(\hat{s}|c)] \\
&= \mathbb{E}_{p_\theta(\hat{s}|c)}[(d(s, \hat{s}) - b)\nabla_\theta \log p_\theta(\hat{s}|c)]
\end{aligned}
$$

where the last line follows from $\mathbb{E}_{p_\theta(\hat{s}|c)}[\nabla_\theta \log p_\theta(\hat{s}|c)] = 0$. With a single-sample Monte Carlo approximation, we have

$$
\nabla_\theta \mathcal{L}_{\mathrm{comp}}(\theta) \approx [d(s, \hat{s}) - b]\nabla_\theta \log p_\theta(\hat{s}|c)
$$

## C    Training details

**Hyperparameters.**  Table 11 and Table 12 shows the hyperparameters of baselines and ESM2-650M, respectively.  For baselines, we follow the training settings in their original papers (Jin et al., 2021; Kong et al., 2022), and for ESM2-650M, we limit the batch size with a maximum token length of 6000 and train for 30 epochs. We set hyperparameter $\alpha = 0.2$ in our sequence objective and use rank 2 for weights $W_q, W_k, W_v$, and $W_o$ in the multi-head attention module for LoRA fine-tuning.

**Implementation for LM-Design.**  In our paper, we adopt LM-Design (Zheng et al., 2023) for antibody design. LM-Design is originally a pLM-based protein design model $p_\theta(\boldsymbol{s}|\boldsymbol{x})$ where $\boldsymbol{x}$ is the protein structure and $\boldsymbol{s}$ is the corresponding sequence that folds into this structure.  For antibody design tasks, we maintain the original architecture, yet use antigen structure as $\boldsymbol{x}$ and initialize $\boldsymbol{s}$ with framework/antigen sequence information.

**Code.**  Our implementation is built upon `https://github.com/facebookresearch/esm`, `https://github.com/BytedProtein/ByProt/tree/main`, `https://github.com/wengong-jin/RefineGNN`, and `https://github.com/THUNLP-MT/MEAN/tree/main`. We deeply appreciate the authors (Lin et al., 2023; Zheng et al., 2023; Jin et al., 2021; Kong et al., 2022) for their contributions to our project.

**Machine specs.**  All models were trained on a machine with 48 CPU cores and 8 NVIDIA Geforce RTX 3090. We have used 1-3 GPUs for all experiments.

Table 11: Hyperparameters for baselines.

|  | LSTM | AR-GNN | RefineGNN | MEAN |
|---|---|---|---|---|
| Vocab size |  |  | 25 |  |
| Dropout |  |  | 0.1 |  |
| Hidden dim |  | 256 |  | 128 |
| Number of layers | 4 | 4 | 4 | 3 |
| K neighbors | - | 9 | 9 | - |
| Block size | - | - | 4 | - |
| Number of RBF kernels | - | 16 | 16 | - |
| Embed dim | - | - | - | 64 |
| Alpha | - | - | - | 0.8 |
| Number of iterations | - | - | - | 3 |
| Optimizer |  |  | Adam |  |
| Learning rate |  |  | 0.001 |  |

Table 12: Hyperparameters for ESM2-650M used as the sequence design model.

|  | esm2_t33_650M_UR50D |
|---|---|
| Vocab size | 33 |
| Embed dim | 1280 |
| Number of layers | 33 |
| Optimizer | AdamW |
| Learning rate | 0.001 |

## D Mitigating exposure bias

As stated in Section 4.3, we address exposure bias by using the generated sequence as the input to the structure prediction model during training. We here demonstrate the benefit of this approach by comparing the performance of the structure prediction model trained with generated sequence (denoted $\hat{s}$) and ground-truth sequence (denoted $s$). Specifically, we train the structure prediction model on SAbDab (fold 0) and RAbD benchmark, then compare their RMSD and TM-score.

Table 13: RMSD and TM-score of structure prediction model trained with generated sequence as input ($\hat{s}$) versus ground-truth sequence as input ($s$). Training the structure prediction model with the generated sequence outperforms its counterpart.

|  | CDR-H1 (SAbDab) | | CDR-H2 (SAbDab) | | CDR-H3 (SAbDab) | | CDR-H3 (RAbD) | |
|---|---|---|---|---|---|---|---|---|
|  | RMSD($\downarrow$) | TM-score($\uparrow$) | RMSD($\downarrow$) | TM-score($\uparrow$) | RMSD($\downarrow$) | TM-score($\uparrow$) | RMSD($\downarrow$) | TM-score($\uparrow$) |
| $\hat{s}$ | **0.7511** | **0.9901** | **0.7993** | **0.9910** | **2.2094** | **0.9748** | **1.78** | **0.9825** |
| $s$ | 0.7077 | 0.9892 | 0.8825 | 0.9895 | 2.4743 | 0.9695 | 1.86 | 0.9789 |

## E Data statistics for SAbDab

In Table 14, we detail the number of antibodies in each fold for 10-fold cross-validation in Section 5.1. Following Kong et al. (2022), $\forall i = 0, \ldots, 9$, we let fold $i$ be the test set, fold $i-1$ be the validation set, and the remaining folds to be the train set.

Table 14: Number of antibodies in each fold for Sequence and Structure Modeling (Section 5.1)

|  | CDR-H1 | CDR-H2 | CDR-H3 |
|---|---|---|---|
| fold 0 | 539 | 425 | 390 |
| fold 1 | 529 | 366 | 444 |
| fold 2 | 269 | 363 | 369 |
| fold 3 | 391 | 380 | 389 |
| fold 4 | 305 | 392 | 420 |
| fold 5 | 523 | 381 | 413 |
| fold 6 | 294 | 409 | 403 |
| fold 7 | 310 | 369 | 327 |
| fold 8 | 472 | 489 | 355 |
| fold 9 | 345 | 403 | 467 |
| **Total** | 3977 | 3977 | 3977 |

# F  ITA algorithm for affinity optimization

**Criterion for validity.** Following (Jin et al., 2021), we force all generated sequences to satisfy the following constraints: (1) net charge must be in the range $[-2.0, 2.0]$, (2) sequence must not contain N-X-S/T motif, (3) a token should not repeat more than five times, (4) perplexity of the sequence must be below 10.

Algorithm 3 details the ITA algorithm in Section 5.3, where we have followed Kong et al. (2022) for the implementation.

---

**Algorithm 3** ITA algorithm for antibody affinity optimization

---

**Input:** SKEMPI V2.0 antibody-antigen complex dataset $\mathcal{D}$, pre-trained $p_\theta(\boldsymbol{s}, \boldsymbol{x}|\boldsymbol{c})$, top-$k$ candidates to maintain

1: Initialize $\mathcal{Q} \leftarrow \mathcal{D}$
2: **for** $t = 1, \ldots, T$ **do**
3:     **for** $(\boldsymbol{s}, \boldsymbol{x}, \boldsymbol{c}) \in \mathcal{D}$ **do**
4:         Initialize $\mathcal{C} \leftarrow \emptyset$
5:         **for** $i = 1, \ldots, M$ **do**
6:             $(\hat{\boldsymbol{s}}_i, \hat{\boldsymbol{x}}_i) \sim p_\theta(\boldsymbol{s}, \boldsymbol{x}|\boldsymbol{c})$
7:             **if** valid$(\hat{\boldsymbol{s}}_i, \hat{\boldsymbol{x}}_i)$ and $f((\boldsymbol{s}, \boldsymbol{x}, \boldsymbol{c}), (\hat{\boldsymbol{s}}_i, \hat{\boldsymbol{x}}_i, \boldsymbol{c})) < 0$ **then**
8:                 $\mathcal{C} \leftarrow \mathcal{C} \cup \{((\hat{\boldsymbol{s}}_i, \hat{\boldsymbol{x}}_i, \boldsymbol{c}), f((\boldsymbol{s}, \boldsymbol{x}, \boldsymbol{c}), (\hat{\boldsymbol{s}}_i, \hat{\boldsymbol{x}}_i, \boldsymbol{c})))\}$
9:             **end if**
10:        **end for**
11:        $\mathcal{Q}[(\boldsymbol{s}, \boldsymbol{x}, \boldsymbol{c})] \leftarrow \mathcal{Q}[(\boldsymbol{s}, \boldsymbol{x}, \boldsymbol{c})] \cup \mathcal{C}$
12:        Select top-$k$ elements in $\mathcal{Q}[(\boldsymbol{s}, \boldsymbol{x}, \boldsymbol{c})]$ by $f(\cdot, \cdot)$
13:    **end for**
14:    **for** $n = 1, \ldots, N$ **do**
15:        Select batch $\mathcal{B}$ from $\mathcal{Q}$
16:        Update $\theta \leftarrow \theta - \eta \nabla_\theta \mathcal{L}(\mathcal{B}, \theta)$
17:    **end for**
18: **end for**

---

# G   Effect of sample recycling on sequence quality

A common NLP approach for reducing token repetition of non-autoregressive models is recycling tokens for $T$ iterations (Ghazvininejad et al., 2019; Zhou et al., 2019). We here demonstrate that this approach is not effective for antibody design.

**Results.** As shown in Figure 6(a), sample recycling does not reduce token repetition of non-autoregressive models on antibody modeling tasks. It even increases $p_{\text{rep}}$ for CDR-H3 from about 15% to 20%. Also, Figure 6(b) shows that sample recycling decreases AARs for all CDR regions. Contrary to this approach, our composition-based objective effectively reduces $p_{\text{rep}}$ while maintaining high AAR, as demonstrated in Section 5.

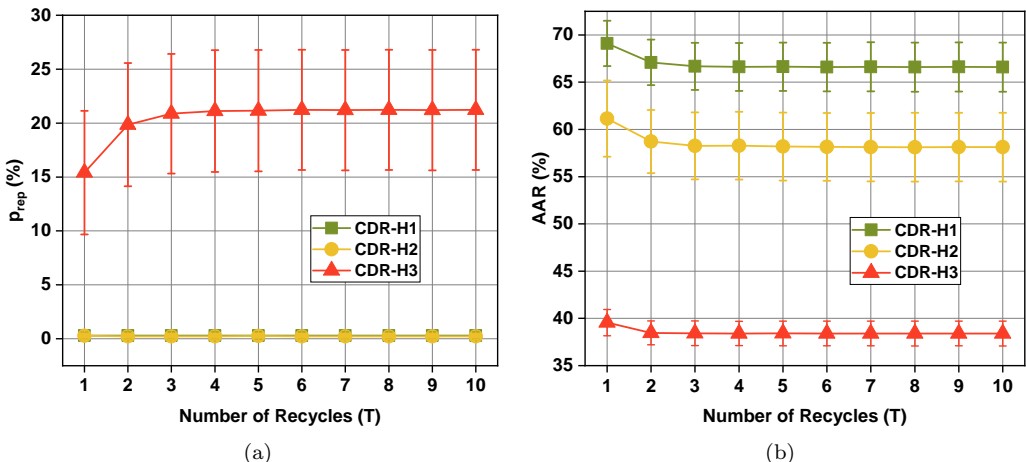

Figure 6: **Effect of sample recycling on $p_{\text{rep}}$ and AAR.** (a) Sample recycling does not reduce token repetition for antibody design tasks. (b) Sample recycling even reduces AAR across all CDRs.

# H   Non-autoregressive (NAR) vs. Autoregressive (AR) models

We discuss why non-autoregressive (NAR) models often outperform autoregressive (AR) models in biological tasks like protein design (Gao et al., 2022; Zheng et al., 2023), peptide sequencing (Zhang et al., 2024), and antibody design (Kong et al., 2022; Verma et al., 2023). Although the reasons for this performance gap are not fully understood, one possible explanation is that the left-to-right inductive bias of AR models conflicts with the structure of biological sequences. In particular, interactions within biological sequences rely on three-dimensional spatial relationships rather than the linear sequence order imposed by AR models. This intrinsic mismatch likely contributes to the weaker performance of AR models in these tasks, aligning with insights discussed in Zhang et al. (2024).

