# OpenReview forum: "Decoupled Sequence and Structure Generation for Realistic Antibody Design"
_TMLR — Accepted by TMLR_

### Review · Reviewer_LHw5 · 2024-10-05

**Summary Of Contributions:**

This paper proposed to decouple sequence and structure generation in antibody design task, and introduced a composition-base objective in the training process to avoid overly repetitive tokens. Using these two approaches, this method can achieve SOTA performance in multiple tasks including antibody design, affinity optimization, protein design and etc.

**Audience:**

Yes

**Claims And Evidence:**

Yes

**Requested Changes:**

1. Missing reference. Under Table 2 on page 7, "Further justifications and discussions on evaluation metrics are in ??"
2. Under Equation 2 on page 6, "$\hat{x} $.... is the predicted sequence", $\hat{x}$ should be the predicted structure given the predicted sequence $\hat{s}$
3. Missing legend for ASSD in Figure 4(b)

**Strengths And Weaknesses:**

Strengths:
1. This paper proposed a decoupled approach to separate sequence and structure design in antibody sequence-structure codesign task. The benefit of this approach is it allows more model architectures for each task and the use of large sequence database in the sequence design step.
2. The authors identified the generation of overly repetitive sequences in non-autoregressive protein design models, which is not desired in practical drug discovery. They propose to use a composition-based objective to regularize the generated sequence which maximize the similarity between target and the predicted sequences.
3. The proposed method is evaluated using multiple benchmarks and experimental settings to verify its performance.

Weakness:
1. The proposed decoupling approach is overly simple, which firstly uses a sequence generator to generate protein sequence, and then a structure predictor is used to predict structure from the generated sequence. This two models are trained separately.
2. Due to the decoupling and the lack of loss propagation between the sequence generator and structure predictor, it is not possible to conditionally design an antibody sequence given an antigen structure, which is the most practical task in structure-based antibody design.
3. In Table 1-3, only CDR-H3 shows significant improvement in $p_{rep}$ using the proposed model. In the case of CDR-H2 and CDR-H1, the baseline models does not show a significant problem of generating repetitive sequences. Why is this?
3. What does the color bar mean in Figure 4?

---

> ### Author Response · Authors · 2024-11-14
> **Response to Reviewer LHw5**
>
> Dear Reviewer LHw5,
>
> We deeply appreciate your thorough review and constructive comments on our manuscript. We have addressed the requested changes (1) and (2) through revisions in the manuscript, which are marked in blue. Regarding (3), this is not ASSD (i.e., sequence-structure decoupling) but rather LM-Design trained with our proposed MLE + RL objective to reduce token repetition. We have modified the legend in the revised manuscript to avoid confusion. Below, we address the reviewer's concerns in detail.
>
> ---
>
> ### Weaknesses
>
> **W1. The proposed approach is overly simple and the two models (i.e., sequence generator and structure predictor) are trained separately.**
>
> We believe that simplicity of our approach is a strength rather than a weakness. Also, as the reviewer mentioned, such two-stage training scheme allows the use of large-scale sequence database for the sequence design step.
>
> **W2. The decoupling prevents designing antibody sequence conditioned on antigen structure.**
>
> We would like to emphasize that recent wet-lab experiments [1] have shown that pre-trained protein language models like ESM can design functional antibody sequences with high binding affinity even without geometric modeling. Our work builds on this empirical finding, suggesting that ESM has implicitly learned structural information, as evidenced in [1] and [2].
>
> Nevertheless, if desired, our sequence generation stage can incorporate structural information as input. This is demonstrated in our adapted implementation of LM-Design, which, however, shows inferior performance compared to our approach. We suspect this may be due to direct architectural modifications to ESM, possibly disrupting the evolutionary information learned from large-scale protein sequence training.
>
> We have revised the Discussion section to clarify this.
>
> ---
>
> ### Questions
>
> **Q1. In Table 1-3, shows that baseline models do not show significant problem in generating repetitive sequences for CDR-H1 and CDR-H2 (in contrast to CDR-H3). Why is this?**
>
> CDR-H1 and CDR-H2 are shorter and less diverse than CDR-H3 [3, 4], allowing models to learn the ground-truth sequences more easily, which results in less repetition.
>
> **Q2. What is the meaning of color bar in Figure 4?**
>
> Thank you for pointing out the possible ambiguity in Figure 4. The color bar represents the value of $\alpha$, which controls the regularization for token repetition in our sequence loss function of Equation 1. In the figure, higher values of $\alpha$ are shown in red, while lower values are in green. This visual illustrates that higher $\alpha$ values more effectively suppress token repetition with a trade-off in AAR. We have clarified this in the caption of Figure 4.
>
> ---
>
> ### References
>
> [1] Hie, Brian L., et al. "Efficient evolution of human antibodies from general protein language models." Nature Biotechnology 2024.
> [2] Lin, Zeming, et al. "Evolutionary-scale prediction of atomic-level protein structure with a language model." Science 2023.
> [3] Morea et al., "Conformations of the third hypervariable region in the VH domain of immunoglobulins." Journal of Molecular Biology 1998.
> [4] Dondelinger et al., "Understanding the Significance and Implications of Antibody Numbering and Antigen-Binding Surface/Residue Definition." Frontiers in Immunology 2018.

---

### Review · Reviewer_pZoP · 2024-10-08

**Summary Of Contributions:**

This paper proposes a two-stage framework, called ASSD, for antibody sequence and structure design. ASSD decouples the joint generation of sequence and structure into a sequence design step and a structure prediction step. The paper also proposes a composition distance loss to penalize the excessive repetition brought by the regular MLE loss. Experiments show ASSD's superior performance over existing AR (autoregressive) and NAR (non-autoregressive) models, and mitigated repetition compared to existing NAR methods.

**Audience:**

Yes

**Claims And Evidence:**

Yes

**Requested Changes:**

- There is a missing reference at the end of the first paragraph at page 7.

**Strengths And Weaknesses:**

Strengths:
- The paper is clear and well written.
- The paper presents convincing experimental results across four benchmarks with multiple metrics. The ablation studies address potential concerns.
- The proposed two-stage framework and composition-based loss is simple to implement.

Weaknesses:
- The paper would benefit from a brief explanation of how non-autoregressive models can outperform their autoregressive counterparts. This seems counterintuitive, as predicting each amino acid should ideally be conditioned on other amino acids.
- Given that the proposed model predicts each amino acid independently, the composition-based loss may not effectively teach the model to avoid repetition. It's possible that the loss simply penalizes large output values in the prediction head rather than encouraging more nuanced sequence learning.

---

> ### Author Response · Authors · 2024-11-14
> **Response to Reviewer pZoP**
>
> Dear Reviewer pZoP,
>
> We deeply appreciate your thorough review and constructive comments on our manuscript. We have incorporated the requested change in our revision, which are marked in blue. Below, we address the concerns in detail.
>
> ---
> ### Question
>
> **Q. Why does non-autoregressive (NAR) models outperform autoregressive (AR) counterparts?**
>
> Non-autoregressive (NAR) models often outperform autoregressive (AR) models in biological-related tasks such as protein design [1,2], peptide sequencing [3], and antibody design [4], though the reasons are not fully understood. A possible explanation is that the left-to-right inductive bias in AR models conflicts with the structure of biological sequences, where token interactions depend on spatial relationships in a 3D structure rather than a linear sequence. This perspective aligns with the explanation provided in [3]. We have revised our manuscript to include this discussion (Discussions, Appendix H).
>
>
> ---
> ### Weakness
>
> **W. The composition-based loss may not effectively teach to avoid repetition and simply penalize large output values in the prediction head.**
>
> We appreciate the reviewer's insightful comment. However, the composition-based loss does more than simply penalizing large output values; it encourages alignment with the natural distribution of CDR sequences and addresses repetition at the level of sequence composition rather than individiual token/head magnitude. Our experiments (Figure 4) confirm the effectiveness of our loss in reducing token repetition.
>
> ---
>
> ### References
>
> [1] Gao et al., PiFold: Toward effective and efficient protein inverse folding, ICLR 23
> [2] Zheng et al., Structure-informed Language Models are Protein Designers, ICML 2023
> [3] Zhang et al., π-PrimeNovo: An Accurate and Efficient Non-Autoregressive Deep Learning Model for De Novo Peptide Sequencing, 2024.
> [4] Verma et al., AbODE: Ab Initio Antibody Design using Conjoined ODEs, ICML 2023

---

> > ### Comment · Reviewer_pZoP · 2024-11-29
> >
> > Thank you for including the discussion about why NARs beats ARs.
> >
> > I am still concerned by the composition-based loss. We did observe the reduced token repetition, however, this might be achieved by penalizing large output values in the prediction head so that no token would be dominant in the output. This is likely because the proposed model predicts each amino acid independently and is not able to actually learn the nuances in sequences. If this is indeed the case, the loss constraints the model's capability to learn certain sequences, for example, where some tokens are dominant.

---

> ### Author Response · Authors · 2024-11-30
>
> **Q. Composition-based loss may reduce token repetition by penalizing large output values in the prediction head so that no token is dominant in the output. (This is likely because the proposed model predicts each amino acid independently and is not able to actually learn the nuances in sequences). If this is indeed the case, the loss contraints the model's capability to learn sequences with dominant tokens.**
>
> We appreciate your follow-up question. We would like to clarify that the composition-based loss does not constrain the model’s ability to learn sequences with dominant tokens. Instead, the loss ensures that the model captures the true distribution of amino acids in the ground-truth sequence. For instance, if the ground-truth sequence is `ARGYYYYY`, the composition loss is minimized only when the model correctly predicts 5 Ys (and 1 A, R, G each), ensuring that both over-prediction *and under-prediction* of dominant tokens are penalized. Conversely, if the model outputs `ARGYDAKS` -- a sequence with no dominant token --  the composition-based loss penalizes this because the predicted composition diverges from the ground truth. This differs from simply penalizing large output values to avoid token repetition, which fails to penalize such under-prediction of dominant tokens.
>
> If your concerns persist, we would be happy to provide additional clarifications. Specifically, could you elaborate on what you mean by the "nuances" of sequences? This will help us address your concerns more directly.

---

### Review · Reviewer_iqzZ · 2024-11-05

**Summary Of Contributions:**

The paper presents ASSD, an antibody loop design framework that decouples sequence design and structure prediction. Specifically, it is trained to only design sequences first and use a separate model for loop conformation prediction. The structure information is hidden from the entire sequence design process. The authors claim this approach allows the use of the most powerful existing models in each individual design phase, such as ESM.
The model demonstrates better performance compared to a selected list of co-design baselines, both AR and non-AR, on metrics like RMSD and Sequence Recovery.  The paper provides ablation study where it modified three previous codesign model into the decoupling framework and showed that it leads to better performance. Furthermore, the paper introduces a composition regularization terms to address the issue of amino acid repetition in one-step non-autoregressive models.

**Audience:**

Yes

**Broader Impact Concerns:**

No Ethical concerns this reviewer can find.

**Claims And Evidence:**

No

**Requested Changes:**

The biggest suggestion is to include more discussion and comparison with most recent co-design approach, especially diffusion/flow matching based ones. They should be considered non-AR as well, and will likely to be immune from repeating aa type problem.
The authors may consider including the reported metrics from AbDiffuser, AbODE, DiffAb and AbDPO, and discuss whether proposed approach can be used on those methods as well.
The authors could also consider Paired-OAS dataset as an additional conditional Ab design task, which is larger than RabD.

The reported metrics in Table 7 ablation study showed exactly the same AAR for the three modified methods, is this an error?

There are several typos and mislabels, e.g. there is a missing reference in page 7 paragraph 1, and the legend in Figure 4(b) should say ASSO instead of LM-Design.

**Strengths And Weaknesses:**

Strength:

Overall the paper introduced an interesting attempt to decouple sequence design completely from structure design. Although counter-intuitive, the paper demonstrates improvement from adopting the decoupled approach to existing models. The experiments covered a wide range of ab design tasks and demonstrated strong results compared to the selected baselines.
The discussion on mode collapse in MLE training of one-step non-autoregressive training that assumes positional independence makes sense, and the proposed method should be effective in diversifying the aa type composition.

Weakness:

The analysis of Non-AR model might be incomprehensive, as it mostly talks about a one-step decoding scenario and ignores methods like diffusion and flow matching, which are state-of-the-art methods on a lot of the biological molecule design problems. The observations on the failure case on MLE is mostly caused by one-step decoding, which is not suitable for learning joint distribution. This issue could be largely alleviated in diffusion as it allows every position to cross-attend with the rest to better capture the joint distribution across the sequence. The proposed regularization loss may not lead to the best convergence to joint distribution as well, as it only tries to mimic the overall amino acid type distribution as opposed to the sequence level profile. Why not use sequence level distance between generated and true sequence?

The biggest concern is the lack of comparisons to recent co-design methods. The baselines selected for the experiments are rather basic and non-sota. The authors claim that they do not compare to AbDiffuser or AbODE because they cannot access their code. However, multiple of the experiments (e.g. on SabDab and RabD) are rather standard, and there is already reported metrics in AbDiffuser/AbODE paper that can be used for benchmarking. Notably, despite the proposed method outperforming their selected baselines, the results are inferior to AbDiffuser/AdODE. E.g., AbDiffuser reported an AAR of 79.09% and 1.12RMSD on SabDab CDR-H1, and AbODE reported 70.5%AAR and 0.65 RMSD, whereas ASSD’s AAR is 69.4%, and RMSD 0.8. Similar data exists for RabD as well.

---

> ### Author Response · Authors · 2024-11-14
> **Response to Reviewer iqzZ**
>
> Dear Reviewer iqzZ,
>
> We deeply appreciate your thorough review and constructive comments on our manuscript. In the revised manuscript, we have marked the changes in blue. In what follows, we address the concerns one-by-one below.
>
> ---
>
>  ### Weaknesses
>
> **W1. Analysis of token repetition problem for Non-AR model is incomprehensive, as it excludes diffusion-based models.**
>
> While we resonate with your concern, our focus on one-/few-step decoding Non-AR approaches are still meaningful since the methods are popular in many biological tasks due to efficient training on large-scale databases and/or fast inference speeds (e.g., [1,2]). We have revised our manuscript to clarify this focus.
>
> **W2. Proposed regularization loss may not be optimal. Why not use sequence level distance?**
>
> We want to clarify that our proposed approach can allow any metric between sequences $d(\mathbf{s},\mathbf{\hat{s}})$; however, we chose cosine similarity between composition vectors for its simplicity and alignment with existing NLP literature. We apologize for this confusion and revised Section 4.2 to make this clearer. Also, if you could point out specific sequence-level distance of interest, we will be happy to run the experiments.
>
> **W3/C1. Comparison with the numbers reported by AbDiffuser/AbODE on standard experiments is possible.**
>
> To alleviate your concern, we provide comparison with AbDiffuser as follows:
>
> |                           | CDR-H1    |          | CDR-H2    |          | CDR-H3    |          |
> |---------------------------|-----------|----------|-----------|----------|-----------|----------|
> |                           | AAR       | RMSD     | AAR       | RMSD     | AAR       | RMSD     |
> | AbDiffuser                | 79.09     | 1.12     | 72.33     | 0.995    | 36.14     | 2.921    |
> | AbDiffuser (side-chains)  | 76.3      | 1.584    | 65.72     | 1.449    | 34.1      | 3.346    |
> | AbDiffuser (τ=0.75)       | 79.76     | 1.083    | 72.96     | 0.95     | 36.53     | 2.805    |
> | AbDiffuser (s.c., τ=0.75) | 76.06     | 1.713    | 66.38     | 1.512    | 34.71     | 3.214    |
> | AbDiffuser (τ=0.01)       | **81.11** | 1.075    | **74.27** | 0.946    | 37.27     | 2.795    |
> | AbDiffuser (s.c., τ=0.01) | 75.36     | 2.463    | 66.89     | 2.01     | 35.56     | 3.124    |
> | ASSD                      | 69.41     | **0.80** | 62.1      | **0.73** | **39.56** | **2.21** |
>
> One can observe that our method is comparable to AbDiffuser (better for four out of six metrics). Please note that our focus is not on achieving state-of-the-art results, but on proposing an idea that generally improves existing joint prediction models.
>
> Unfortunately, it is hard to fairly compare with AbODE due to mismatch in evaluation metrics. Upon inspecting the AbODE's evaluation code, we found that AbODE predicts sequences of length $n-2$ for a CDR sequence of length $n$, which does not align with our setup. We provide evidence of this issue in a demonstration notebook ([link](https://anonymous.4open.science/r/AbODE_tmlr-0228/demonstration.ipynb)).
>
> ---
>
> ### Requested changes
>
> **C2. Use Paired-OAS dataset, which is larger than the RAbD dataset.**
>
> Thank you for the helpful suggestion. However, the focus of our paper is on antibody sequence-structure co-design, while the Paired-OAS is a sequence database that does not contain structural information.
>
> **C3. Possible repeated entries in Table 7.**
>
> We apologize for the confusion. The entries are repeated since the decoupled variants all use the same pLM as the sequence designer and thus share the same results for sequence. We have clarified this in our revised manuscript (Table 7 caption, Section 7.1 Results).
>
> **C4. Missing reference in page 7**
>
> We appreciate your careful review and have made corresponding changes.
>
> **C5. Mislabel for Figure 4b (should be ASSD instead of LM-Design)**
>
> We apologize for the confusion. This is not ASSD (i.e., sequence-structure decoupling), but LM-Design trained with MLE+RL, which was proposed in our paper, as explained in the caption. We have modified the legend of Figure 4b to make this more clear.
>
> ---
>
> ### References
> [1] Zheng et al., Structure-informed Language Models are Protein Designers, ICML 2023
> [2] Zhang et al., π-PrimeNovo: An Accurate and Efficient Non-Autoregressive Deep Learning Model for De Novo Peptide Sequencing, 2024.

---

### Decision · Action_Editor_PipY · 2024-12-11

**Recommendation:** Accept as is

**Comment:**

The paper proposes a decoupling approach to separate sequence and structure design. The paper also introduces a composition-based objective for antibody sequence generation that resolves the token repetition problem

The key claims are that (i) the experiments demonstrate that the proposed sequence-structure decoupling approach improves performance in a range of antibody design experiments; and (ii) the introduced training algorithm effectively prevents excessive token repetitions.

While the reviewers expressed some minor concerns in their recommendations regarding the selection of baselines and the adopted sequence design methods, and whether the proposed composition loss was particularly effective, there was consensus that the experiments were sufficient to support the claims. There were requests to include additional baselines during the review process and the authors satisfactorily addressed those requests. Insisting on the inclusion of additional, unspecified baselines (“strengthening of the baseline selection”) after the completion of the review process is unreasonable. The authors provided satisfactory responses to the other criticisms of the reviewers concerning simplicity of the technique and the effectiveness and suitability of the composition loss.

Audience: A subgroup of TMLR’s audience is interested and conducts research into antibody design. The presented results indicate that a simple method can outperform some recent techniques. This counterintuitive finding is likely to be of interest for the audience, and the introduced method can potentially serve as an effective baseline for future research.

Certifications: The paper does not meet the bar for any certifications.

**Audience:**

Some individuals would be interested in knowing the findings of the paper.

**Claims And Evidence:**

The claims in the submission are supported by accurate, convincing and clear evidence.

---

> ### Author Response · Authors · 2025-01-02
>
> Dear Action Editor,
>
> We sincerely appreciate your time and effort in the reviewing process and your support of our work. We have submitted the camera-ready version of our paper.
>
> Thank you again for your support.
>
> Best regards,
> The Authors